# Ytterbium-Doped Lead–Halide Perovskite Nanocrystals: Synthesis, Near-Infrared Emission, and Open-Source Machine Learning Model for Prediction of Optical Properties

**DOI:** 10.3390/nano13040744

**Published:** 2023-02-16

**Authors:** Yuliya A. Timkina, Vladislav S. Tuchin, Aleksandr P. Litvin, Elena V. Ushakova, Andrey L. Rogach

**Affiliations:** 1International Research and Education Centre for Physics of Nanostructures, ITMO University, Saint Petersburg 197101, Russia; 2Laboratory of Quantum Processes and Measurements, ITMO University, Saint Petersburg 197101, Russia; 3Department of Materials Science and Engineering & Centre for Functional Photonics (CFP), City University of Hong Kong, Hong Kong SAR 999077, China

**Keywords:** lead–halide perovskite, doping, synthesis, quantum cutting, quantum yield, machine learning, multiple regression

## Abstract

Lead–halide perovskite nanocrystals are an attractive class of materials since they can be easily fabricated, their optical properties can be tuned all over the visible spectral range, and they possess high emission quantum yields and narrow photoluminescence linewidths. Doping perovskites with lanthanides is one of the ways to widen the spectral range of their emission, making them attractive for further applications. Herein, we summarize the recent progress in the synthesis of ytterbium-doped perovskite nanocrystals in terms of the varying synthesis parameters such as temperature, ligand molar ratio, ytterbium precursor type, and dopant content. We further consider the dependence of morphology (size and ytterbium content) and optical parameters (photoluminescence quantum yield in visible and near-infrared spectral ranges) on the synthesis parameters. The developed open-source code approximates those dependencies as multiple-parameter linear regression and allows us to estimate the value of the photoluminescence quantum yield from the parameters of the perovskite synthesis. Further use and promotion of an open-source database will expand the possibilities of the developed code to predict the synthesis protocols for doped perovskite nanocrystals.

## 1. Introduction

During the last decade, lead–halide perovskite nanocrystals (pNCs) arose as luminescent materials with attractive optical properties. Their intrinsic defect tolerance [1], combined with the development of methods for effective surface passivation bring the opportunity to obtain pNCs with a photoluminescence quantum yield (PL QY) close to unity [2,3]. This fascinating luminous efficiency makes them promising materials for light-emission and light-conversion applications, including light-emitting diodes, lasers, luminescence solar concentrators, and planar solar cells. [4,5,6,7]. The doping of perovskite materials emerges as the perspective direction of expanding their chemical and physical properties, including crystalline structure, emissive properties, energy level structure, and recombination dynamics [8]. For example, doping the CsPbBr_3_ pNCs with divalent cation (Cd^2+^) was shown to improve both their linear and nonlinear optical properties [9]. Modification of perovskite with dual cations (divalent Cd^2+^ and monovalent K^+^) was shown to be very efficient in reducing the defects, immobilizing the halide anions, and preventing ion loss from perovskite during post-annealing process which improved power conversion efficiency of perovskite solar cells [10].

Moreover, high absorption cross-section is another essential characteristic of these materials, expanding their functionality by utilizing them as a sensitizing host. Hence, various ions with their own emission bands can be incorporated into perovskite crystal lattice to be sensitized through efficient light absorption by the perovskite host. Doping pNCs with ytterbium (Yb^3+^) ions is a convenient strategy to realize near-infrared emission (NIR) with a peak at ~980 nm. Engineering the perovskite stoichiometry and the dopant content allows us to obtain either dual-band emitting or predominantly NIR-emitting pNCs. As compared to the classical II-VI semiconductor quantum dots (CdS, CdSe, CdTe, etc.), Pb ions in pNCs have a coordination number of 6 (octahedral coordination) which facilitates doping with lanthanides [11,12]. Strong and tunable absorption of the perovskite host matrix followed by an efficient transfer of the photoexcitation to the dopant site provides a high PL QY of the NIR emission. Strong NIR emission from Yb^3+^-doped pNCs is attractive for the fabrication of down-converters for silicon solar cells [13], luminescent solar concentrators [14], NIR light-emitting diodes [15], dual-band photodetectors [16], and NIR luminescent labels operating under two-photon excitation [17]. In particular, Zhou et al. deposited a layer of Yb^3+^-Ce^3+^ co-doped CsPbCl_1.5_Br_1.5_ pNCs on top of a commercial single-crystal silicon solar cell [13]. Optimization of the thickness of a perovskite layer provided both high transparency in the visible and NIR regions and effective capture of the ultraviolet light which was converted to the Yb^3+^-related NIR radiation. As a result, the power conversion efficiency increased from 18.1% to 21.5%. Luo et al. demonstrated a concept of a quantum cutting luminescent solar concentrator with Yb^3+^-doped CsPbCl_3_ pNC embedded into a polymethyl methacrylate matrix [14]. The prototype of a solar concentrator with the dimensions of 5 cm × 5 cm × 0.2 cm demonstrated an internal quantum efficiency of 118.1 ± 6.7%, 2-fold higher than previous records. Huang et al. deposited Yb^3+^-doped CsPbCl_3_ onto a UV commercial chip to build a NIR-emitting light-emitting diode [15]. The device demonstrated peak external quantum efficiency of 2% and good operational stability. Sun et al. used laterally assembled Yb^3+^-doped CsPbClBr_2_ nanosheets to fabricate a dual-band sensitive photodetector [16]. The device demonstrated the responsivity of 1.96 A·W^−1^ (0.12 mA·W^−1^) and detectivity of 5 × 10^12^ (2.15 × 10^9^) Jones at 440 (980) nm wavelength.

Recently, machine learning algorithms stepped on the scene of nanomaterial synthesis since they can significantly accelerate the optimization of synthetic protocols, and potentially lead to better pathways toward new materials with desired properties [18,19]. In 2021, Srivastava et al. proposed a roadmap for perovskite photovoltaics highlighting several promising research directions: compositional selection, perovskite material synthesis and testing, and photovoltaic device evaluation [20]. In the same year, Tao et al. pointed out the exponential growth of machine-learning strategies in the perovskite field [21]. At the same time, the authors also highlighted the obstacles to future applications of machine learning such as lack of data, establishing and sharing datasets for perovskites, and a better understanding of the chemical and physical meaning of the mathematical models. Thus, the broad availability of mathematical models, including machine learning, is in high demand for further development of doped perovskite fabrication methods.

In this review, we will consider both the synthetic methods and optical properties of Yb-doped pNCs with a focus on the NIR emission from Yb^3+^ ions in the perovskite host matrix. We introduce the quantum cutting phenomenon and shortly review its models proposed in the recent literature. The data on synthetic parameters with dependencies of optical responses for Yb-doped perovskites are summarized and approximated by linear and multiple parameter linear regressions using machine learning methods. We believe that the developed open-source code will help researchers to improve synthetic protocols and develop novel lanthanide-doped perovskite materials.

## 2. Synthesis

Generally, metal–halide perovskites are described by a structural formula ABX_3_, where A is a mono-valent organic/inorganic cation such as methylammonium (MA^+^), formamidinium (FA^+^), Cs^+^, or Rb^+^; B—bi-valent metal cation such as Pb^2+^ and Sn^2+^; and X—halogen anion such as Cl^−^, Br^−^, and I^−^. Thanks to this structure of the crystal lattice, the perovskite materials can be easily doped with trivalent lanthanides (Ln^3+^) via octahedral coordination such as Sm^3+^ [22], Yb^3+^ [23], La^3+^ [24], Nd^3+^ [25], Ce^3+^ [13], Er^3+^ [26], Tb^3+^ [27], Dy^3+^ [23], and Y^3+^ [28]. A schematic illustration of the doping process of the lead–halide perovskite crystal lattice with Yb^3+^ cations is provided in Figure 1.

In recent years, much attention has been paid to the development of reliable methods for the synthesis of doped metal–halide perovskites. Those can be generally divided into single-step methods where doping occurs directly during the synthesis [23,29] (Figure 2a illustrates a prominent example—the so-called “hot-injection method”), and post-synthetic treatment using anion (cation) exchange reactions which result in post-synthetic doping [30,31] (Figure 2b).

Among single-step methods used for the synthesis of doped pNCs, already mentioned hot-injection method (Figure 2a), ligand-assisted reprecipitation (LARP), super-saturated recrystallization [32], and ultrasonic synthesis [27] can be named. The hot-injection method is a fast and widely used approach for the preparation of monodisperse NCs with controllable size and shape via temperature and reaction time. Initially, this method was developed for classical semiconductor quantum dots [33], and then adapted for lead–halide pNCs [34]. The essence of the hot-injection method is that the nanoparticle precursors are first prepared separately and then quickly mixed or injected under certain reaction parameters (Figure 2a). A sharp supersaturation results in the initiation of explosive nucleation of nanoparticles from the so-called monomers. The crystal size can be controlled by reaction kinetics, i.e., injection temperature and rate, reaction time, use of seeds or catalytic sites, presence of reducing or oxidizing agents, and reaction atmosphere [35]. Using this method, it is possible to obtain pNCs with a precisely controlled narrow size distribution and excellent optical properties [36]. Among the disadvantages of this method is the necessity of using rather high temperatures, the difficulty of scaling up, as well as strong sensitivity of the properties of the resulting product on synthetic deviations [37]. Recent protocol [38] has summarized all the details of the hot-injection method for green-emitting CsPbBr_3_ pNCs. Another single-step method to produce doped pNCs is the LARP method [39]. The essence of this method is that when certain precursors and chemical reagents with different solubilities are added to the solution, a sharp saturation will occur, triggering the nucleation and growth of nanoparticles [40]. The advantages of this method are that it is typically conducted at room temperature, and the possibility of scaling up the synthesis [41], while lower stability of the resulting nanocrystals, including sensitivity to polar solvents, are among the disadvantages [40]. To date, only lead-free Eu-doped pNCs were synthesized by LARP [42].

Post-synthetic treatment is another doping method, where the shape and crystal structure of pNCs are preserved. Typically, initially synthesized pNCs are treated with lanthanide salts for the cation exchange under continuous stirring for several minutes (Figure 2b) [43].

For both single-step and post-synthetic treatment, the sources of Yb^3+^ ions for doped perovskites are different salts such as ytterbium acetate (Yb(CH_3_COO)_3_ or Yb(OAc)_3_) [44], nitrate (Yb(NO_3_)_3_) [30], oleate (Yb(C_18_H_34_O_2_)_3_) [15], and chloride (YbCl_3_) [13,45]. For the latter source, an issue of the limited solubility in organic solvents has been mentioned [46].

## 3. Morphology of Yb-Doped Metal–Halide pNCs

When lead–halide pNCs are doped with lanthanide ions such as Yb^3+^, the nanoparticle morphology can change. Since the atomic radius of Yb^3+^ is smaller than the atomic radius of Pb^2+^, such doping results in a decrease in the interplanar distance, which is observed as the shift of the reflection peaks towards larger angles in X-ray diffraction patterns [28,43]. On the other hand, the introduction of Yb^3+^ ions results in the formation of a vacancy at the Pb^2+^ site {V(Pb^2+^)} maintaining charge neutrality as shown in Figure 1, which also leads to a decrease in the interplanar distance in the crystal lattice. At the same time, the bonding energy increases [28,43], which makes doped pNCs more resistant to external conditions. On the other hand, doping with Yb^3+^ ions does not change the cubic shape of pNCs, indicating the preservation of their cubic phase [15]. Moreover, doping of perovskites with lanthanides increases the probability of oxidation of Pb^0^ defects to Pb^2+^, thereby increasing perovskite stability and emission efficiency [47].

## 4. Optical Properties

The ability to choose different lanthanide ions for perovskite doping is a useful tool to realize NIR emission, including a change of the PL band position and the relaxation of charge carriers. Zeng et al. reported NIR emission from lanthanide ions in the perovskite host with PL QY of ~0.8% at 1542 nm, ~0.7% at 986 and 1484 nm, and ~3% at 888, 1064, and 1339 nm for Er^3+^, Ho^3+^, and Nd^3+^ doping, respectively, where NIR emission was achieved via two-step energy transfer from the perovskite host to Mn^2+^ dopant, and then to lanthanide ions [25]. The doping with lanthanide ions also introduces additional energy levels, i.e., defect states, within the perovskite’s bandgap. It was shown by density functional theory (DFT) calculations that Ce^3+^ (lanthanide) doping results in the formation of shallow traps as compared to deep trap state formation in the case of Bi^3+^ (non-lanthanide) doping [48]. Different from the deep traps, shallow traps facilitate an increase in electron density in the conduction band of the perovskite host and, hence, may lead to PL enhancement. Zhou et al. showed that Yb-doping can increase the PL QY from 30 to 50% for CsPbBrI_2_ pNCs [49]. Doping with divalent Yb^2+^ ions can also result in the improvement of the environmental and thermal stability of perovskite materials, as was shown for Yb^2+^-alloyed CsPb_1−x_Yb_x_I_3_ pNCs [50]. The formation of defect states was experimentally shown for pNCs doped with different lanthanide ions in a recent study by Gamelin’s group [51]. Trivalent doping of this type was shown to generate a new and often emissive defect state observed at approx. 50 meV below the perovskite host conduction band, which was independent of the specific lanthanide dopant but varied with perovskite chemical composition [51]. Recently, Yang et al. showed that co-doping with La^3+^ and Yb^3+^ resulted in an increase in total PL QY from 5 to 168% by reducing initial trap states through both La^3+^ and Yb^3+^ down-conversion of perovskite photoexcitation to Yb^3+^ emission via quantum cutting [24].

### Yb^3+^ Emission and Quantum Cutting

In the oxidation state of 3^+^, Yb emission occurs between the energy levels ^2^F_5/2_ and ^2^F_7/2_ (1.26 eV or 980 nm); however, this transition is very weak since it is forbidden in a centrosymmetric system according to the Laporte rule. The ^2^F_5/2_ energy level can be populated via excitation transfer from the perovskite host, increasing the number of charge carriers and thus the NIR PL intensity. The structure Yb^3+^–X^−^–V(Pb^2+^)–X^−^–Yb^3+^ (illustrated in Figure 1) forms a trap state, facilitating the coupling of perovskite host with two Yb^3+^ ions and leading to the fast nonradiative energy transfer from the photoexcited pNC host to those two Yb^3+^ ions simultaneously, which is known as *quantum cutting*. The quantum cutting phenomenon can be considered as an energy transfer down-conversion [52]. In this process, the system containing two types of ions (type I—absorbing and emitting ultraviolet and visible light, type II—lanthanide ion with optical transitions in red and NIR spectral region) is excited by a high-energy photon; the type I ion absorbs this photon and then may relax to the ground state with the emission of a single photon or transfer its energy to the type II ion with its subsequent emission of a photon with less energy. Quantum cutting may occur if an absorbed by the type I ion photon’s energy is twice as large as the energy of the subsequent emission from the type II ion. As the energy absorbed is split into two low-energy emitting photons, the overall PL QY may exceed 100% and even reach the theoretical maximum of 200% [53]. This phenomenon was proposed to explain the extraordinarily high PL QY of Yb-doped and Yb^3+^, Ce^3+^ co-doped pNCs [13]. Due to the great promise of this phenomenon for improving the efficiency of light harvesting devices (the emission energy of ytterbium ions matches perfectly with the absorption band of commercial silicon solar cells) [13,14], the quantum-cutting process in lead–halide pNCs was further studied in detail. Gamelin’s group attributed the high efficiency of NIR emission in Yb-doped CsPbCl_3_ pNCs to a very high rate (in a ps time scale) of a photoexcitation transfer from a perovskite host to two Yb^3+^ ions [46].

To date, there are two different mechanisms proposed for the quantum cutting phenomenon in the lanthanide-doped pNCs. In 2017, Pan et al. proposed a mechanism where a deep defect energy level located almost in the middle of the perovskite host bandgap is involved [23]. Followed by photoexcitation of the perovskite host, electrons relax to the deep defect state with released energy transferred to the first Yb^3+^ ion, and then electrons recombine with the holes in the valence band with energy transfer to the second Yb^3+^ ion (Figure 3a). In 2018, Gamelin’s group inferred that Yb^3+^ doping results in the formation of a shallow lattice defect [46]; followed by the photoexcitation of the perovskite host electrons relax firstly to this shallow defect state and then recombine with holes in the valence band while the released energy is transferred to two Yb^3+^ ions (Figure 3b). DFT calculations conducted by Li et al. further supported the latter mechanism involving shallow defect state formation in Yb-doped CsPbCl_3_ pNCs [54].

The existence of the shallow traps induced by Yb-doping of CsPbCl_3_ pNC was also shown experimentally in [15]. The comparison of the normalized PL spectra in the visible spectral range at different temperatures indicates an increase in the shallow state PL intensity at room temperature (Figure 3c). Together with the similarity of PL excitation spectra in visible and NIR spectral ranges with absorption spectrum, and the Yb-doping induced decrease in the visible PL lifetime from 8.2 to 1.6 ns, this indicates an efficient energy transfer from the shallow trap states in the pNC host to Yb^3+^ ions (inset in Figure 3c).

The quantum cutting effect and the values of NIR PL QY depend on the perovskite host bandgap and, hence, its chemical composition [11,30]. Mir et al. showed that indeed the NIR PL intensity depends on the chemical composition of the pNC (Figure 4a): the 0.6% Yb-doped CsPbCl_3_ pNCs with the largest bandgap showed the highest PL QY. The most intense NIR emission was observed when the bandgap of the pNCs was approximately double (~2.6 eV) of emission energy from the Yb^3+^ ions when X varied from Cl to different ratios of Cl/Br anions in Yb-doped CsPbCl_x_Br_3−x_ pNCs [44]. A similar observation was recently reported for the Yb-doped CsPbCl_x_Br_3−x_ nanosheets (Figure 4b) [16].

## 5. Dependence of the Properties of Yb-Doped Lead–Halide pNCs on Synthesis Parameters

Both optical properties and morphology of doped lead–halide pNCs depend on the synthesis parameters such as type of precursors and ligands, their molar ratio, temperature of synthesis, etc. The available literature data on Yb-doped CsPbX_3_ pNCs produced both by the hot-injection method and by post-synthetic treatment are summarized in Table 1. The extended version of this Table is available on https://github.com/VladislavNexby/ML (accessed on 15 February 2023) as train.csv, where types of precursors are put in different columns; furthermore, the solvent of the visible (Vis) PL band and the position of NIR PL peak are provided. These parameters may facilitate choosing the data of interest, including synthetic, structural, and optical data, for further modeling. In this review, we highlight several parameters for further analysis in terms of pNC size and Yb content, as well as their PL QY, both in NIR and visible (Vis) spectral regions: amount and type of ligands, temperature of the reaction, and Yb-precursor type and its amount in the synthesis. All the dependencies revealed in the discussion later are based on the published data from the papers [13,15,17,23,26,31,43,44,46,55,56,57,58,59,60] and provided in Table 1.

### 5.1. Dependence on the Ligand Type and Its Amount

The type of ligands and their amount in the precursor mixture is known to affect both the shape and size of the produced nanoparticles. Recently, Sun et al. showed that the CsPbCl_3_ nanosheets (100–300 nm in lateral size) were obtained using the octylamine only while using oleylamine resulted in the formation of 8.3 nm cubic nanocrystals [16]. As summarized in Table 1, the major ligands used in the synthesis of cubic CsPbX_3_ pNCs are oleic acid (OlAc) and oleylamine (OlAm). Their amount in the reaction mixture together with the molar ratio affects both the morphology and optical properties of synthesized NCs. Herein, the syntheses which used both OlAm and OlAc were considered for further analysis (Figure 5a–d) [13,15,17,23,26,31,43,44,46,55,56,57,58,59,60].

From Figure 5a, one can see that the size of Yb-doped CsPbX_3_ pNCs decreases with an increase in the molar ratio of [OlAm/OlAc], with linear regression described as D = 13 − 3.2∙[OlAm/OlAc]. The increase in the ratio [OlAm/OlAc] leads to an increase in the measured Yb content in NCs (Figure 5b). It should be noted that for [OlAm/OlAc] equal to 0.5 and 1 the Yb content is almost the same within a 25–75% data interval. From Figure 5c, there is no obvious dependence on [OlAm/OlAc] of PL QY in the visible spectral range (Vis PL QY), as the data demonstrate a vast spread of values. However, the [OlAm/OlAc] ratio affects the Yb-related NIR QY: with an increase in the mean [OlAm/OlAc] value the NIR PL QY decreases (Figure 5d). The last dependence can be described by linear regression in a form: NIR PL QY = 133 − 37.9∙[OlAm/OlAc].

### 5.2. Dependence on Reaction Temperature

It is well known that by reaction temperature one can tune the reaction rate affecting the morphology and resulting optical properties of synthesized NCs. By increasing the reaction temperature from 180 to 280 °C one can obtain larger Yb-doped pNC varies, as was shown in [16]. In [26] it was shown that the maximal NIR PL intensity of such pNCs was observed at 260 °C compared to the values achieved at 200–280 °C. The analysis of the literature data shows that the size of pNCs is almost independent of the reaction temperature (Table 1). However, for the reported synthesis at 200 °C [13,57], there is a smaller spread in the size values (Figure 5e). The measured Yb content weakly depends on temperature: a larger percentage of Yb is observed for syntheses at temperatures above 200 °C as shown in Figure 6f. Vis PL QY is temperature-independent (Figure 5g), while NIR PL QY increases with temperature up to approx. 140% at 240 °C and then decreases to 120% at 260 °C (Figure 5h). At temperatures higher than 260 °C, NIR PL QY decreases which can be caused by concentration quenching [26], which is discussed in the next paragraph.

### 5.3. Dependence on the Yb-Precursor Type and Its Amount

The NIR PL excited via perovskite host can be influenced by the Yb dopant content in the pNC, and the quality of the crystal lattice (amount of crystal defects introduced by doping). As mentioned above, at first, an increase in the Yb content results in a decrease in the crystal defects by forming Yb^3+^–X^−^–V(Pb^2+^)–X^−^–Yb^3+^ structure which facilitates NIR emission, but a further increase in the Yb content results in an impairment of the perovskite crystal structure followed by a decrease in NIR PL QY. It was also shown experimentally that there exists the saturation of dependence of the Yb content on the Yb/Pb precursor molar ratio. Zhao et al. showed that the NIR PL QY increased up to 90% with an increase in Yb content to 10%, and then experienced a decrease upon an increase in Yb content to 30% [43]. Huang et al. showed that at 0.5 (‘reaching a critical value of 0.45′) Yb/Pb precursor molar ratio the Yb content reached the maximum of 7 mol% and then with an increase in Yb/Pb remained the same [15]. Dagnall et al. probed Yb content from 0 to 60% and showed that the maximal radioluminescence intensity of CsPb(Cl_1−x_Br_x_)_3_ pNCs was observed for 5% [59]. A similar trend was observed for lead-free pNCs, with maximum PL QY observed at 3 mol% for Cs_2_AgBiCl_6_ and Cs_2_AgBiBr_6_ [60].

We examined the influence of the type of Yb-precursor on NIR PL QY of the resulting pNCs for six different kinds of Yb salts (Figure 6a), among which the most popular ones are YbCl_3_ and Yb(OAc)_3_ (see Table 1). For all six types of salts, the NIR PL QY remains almost constant (Figure 6a), which indicates that at the synthesis conditions, these Yb salts completely dissociate providing Yb monomers to be involved in pNCs growth. Dependencies of size, Yb content, and PL QYs on the molar ratio of Yb and Pb precursors, [Yb/Pb], are shown in Figure 6b–e, respectively. The size of CsPbX_3_ pNCs varies in the range of 6–16 nm for [Yb/Pb] varied from 0.1 to 3. The measured Yb content shows a large spread of reported values with a maximum at [Yb/Pb] = 0.4 (Figure 6c). Both Vis and NIR PL QYs show a maximum at [Yb/Pb] = 1, which is consistent with the trend for the sample set published in [15]. With an increase in the [Yb/Pb] molar ratio up to 1, NIR PL QY also increases almost linearly; for [Yb/Pb] > 1, the NIR PL QY decreases and saturates at 100–120% for [Yb/Pb] > 3 (Figure 6e).

### 5.4. Multi-Parameter Approximation for NIR PL QY

From the above discussion of the literature data, dependencies of structural and optical properties of Yb-doped pNCs on each synthesis parameter seems to be not that univocal. This inspired us to reveal the multiple linear regression model which could predict the NIR PL QY from the initial synthesis parameters for Yb-doped CsPbCl_3_ based on the data shown in Appendix A [15,23,26,45,46,56,57,58]. Our code was written with Google Colab (Python 3.8.16) which is a hosted Jupyter notebook service. All Colab notebooks are stored in the open-source Jupyter notebook format (.ipynb). The link to our code is: https://github.com/VladislavNexby/ML (accessed on 15 February 2023).

The following parameters were considered: the amount of Pb and Yb salts in mmol, the amount of ligands (OlAm and OlAc) in mL, the reaction temperature in °C, and the reaction time in seconds (Appendix A). Appendix A shows the result of the implementation of the linear regression model in the array of predicted values of NIR QY compared to experimental values taken from the Refs. [15,23,26,45,46,56,57,58]. According to the outcome of modeling, the coefficient of determination (R2) is 0.84, the root-mean-square error (RMSE) is 15.8 and the mean absolute error (MAE) is 11.5. The resulting value of R2 is close to 1, which is a good result, but the values of RMSE and MAE are too large due to the small amount of data. The dependence of NIR PL QY was estimated by multiple linear regression with coefficients as weights: $\mathrm{NIR}\;\mathrm{PL}\;\mathrm{QY}\;\left(\%\right)={\;\beta }_{1}\cdot {\mathrm{Pb}}_{\mathrm{mmol}}+\beta_{2}\cdot {\mathrm{Yb}}_{\mathrm{mmol}}+\beta_{3}\cdot {\mathrm{OlAm}}_{\mathrm{mL}}+\beta_{4}\cdot {\mathrm{OlAc}}_{\mathrm{mL}}+\beta_{5}\cdot \mathrm{temperature}+\beta_{6}\cdot {\mathrm{time}}_{\sec}$. The estimated coefficients $\beta_{1}$–$\beta_{6}$ are given in Appendix A.

## 6. Summary and Outlook

To date, perovskite materials, and in particular all-inorganic lead–halide perovskite nanocrystals, attract a lot of scientific attention due to the wide field of their potential applications. However, these materials are still far from industrial use since they have several drawbacks such as fast oxidation and poor photostability in the UV spectral region. An important target is the development of simple, scalable, and reliable synthetic procedures for producing perovskite materials with well-controlled properties such as bright emission and tunable energy structure. The doping of pNCs with lanthanides, along with expanding the spectral range of their optical transitions to NIR, allows us to overcome some existing issues of perovskite materials, by improving their environmental stability, decreasing the intrinsic defects and, hence, improving the emissive properties in the visible spectral region. Furthermore, doping with Yb^3+^ ions results in an increase in PL QY up to 200% via the efficient energy transfer from the perovskite host and quantum cutting. The literature analysis shows that the process of the quantum cutting depends on the bandgap of the perovskite host which can be varied by a change of X site from Cl to different ratios of Cl/Br anions in CsPbX_3_ pNCs. The analysis of reported data also shows dependencies of NIR PL QY on the synthetic parameters of Yb-doped CsPbX_3_ pNCs such as ligands ratio, reaction temperature, and the molar ratio of Yb and Pb precursors. For instance, a distinct maximum was observed for NIR PL QY at equal molar amounts of Yb and Pb precursors in the synthesis. The optimization of synthetic procedures may lead to a better performance of perovskite-based devices. Obviously, extended use of the existing and emerging instruments of data analysis and machine learning will play an important role in improving the synthetic protocols and developing novel lanthanide-doped perovskite materials.

Based on the literature data, an open-source code for the approximation of NIR PL QY dependence on synthetic parameters was proposed. This model can solve the direct problem of predicting the values of the PL QY in the near-IR region of the spectrum from the synthesis parameters of CsPbCl_3_ pNCs and it can be considered as a base for further improvements, which may be accomplished in several possible ways:(i)Development of a standardized template for data collection and further expansion of the dataset based on available experimental data will improve the accuracy of the model prediction;(ii)Other machine learning models, regularization techniques, and other methods of preprocessing can be employed and compared for better model performance;(iii)Improvement of mathematical models may include such functions as a prediction of maximal values, sorting by type of precursors and/or resulting materials, graphical representation of derived dependencies of optical characteristics on synthesis parameters, and many more.

Properly performed data analysis will facilitate the optimization of existing synthesis methods together with the possibility to predict the optical properties of doped perovskite materials by setting the synthesis parameters. Thus, the employment of machine learning methods offers a great potential for improvement of the doped perovskite materials in particular and plenty of novel nanostructured materials in general.

## Figures and Tables

**Figure 1 nanomaterials-13-00744-f001:**
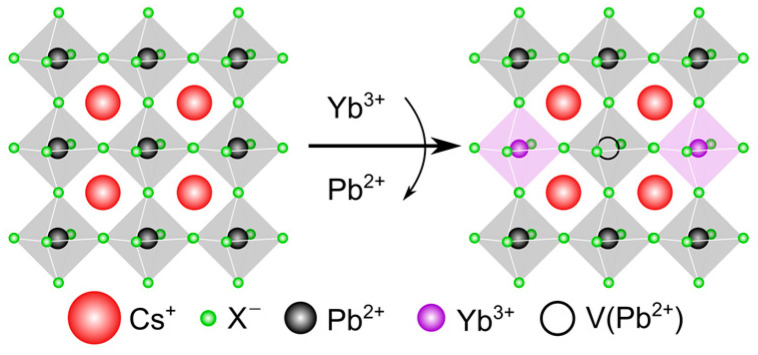
Schematic illustration of the doping process of the lead–halide perovskite crystal lattice with Yb^3+^ cations.

**Figure 2 nanomaterials-13-00744-f002:**
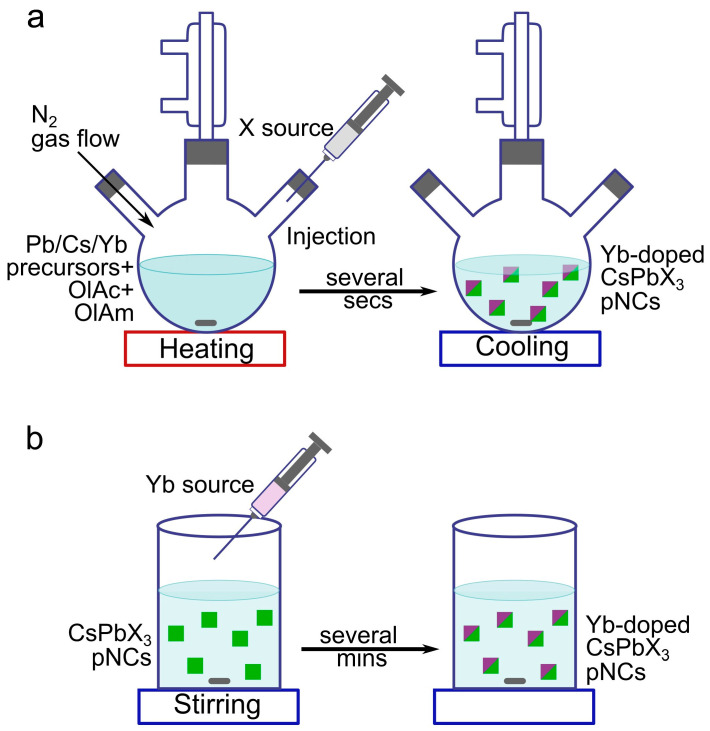
Illustration of the hot-injection method (**a**) and post-synthetic treatment (**b**) to obtain Yb-doped pNCs.

**Figure 3 nanomaterials-13-00744-f003:**
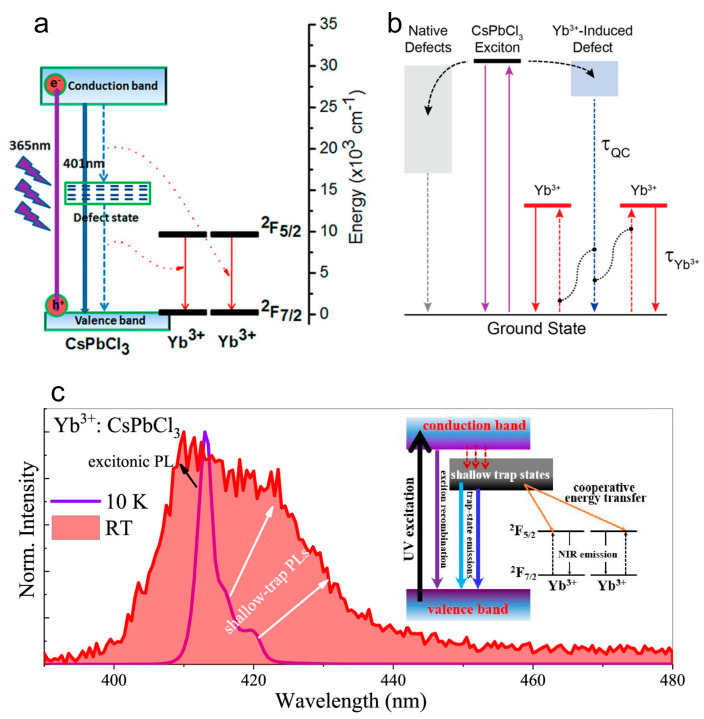
Two models of the energy structure of Yb-doped CsPbCl_3_ perovskite involving (**a**) deep defect state {Adapted with permission from [23]. Copyright 2017 American Chemical Society}, and (**b**) shallow defect state {Adapted with permission from [46]. Copyright 2018 American Chemical Society}, which are used for explanation of the quantum cutting effect. (**c**) Normalized PL spectra of Yb-doped CsPbCl_3_ perovskite in the wavelength range of 380–480 nm. Inset is the proposed pNC-sensitized Yb^3+^ PL quantum cutting mechanism involving shallow trap state-assisted cooperative energy transfer. Adapted with permission from [15]. {Copyright 2021 American Chemical Society}.

**Figure 4 nanomaterials-13-00744-f004:**
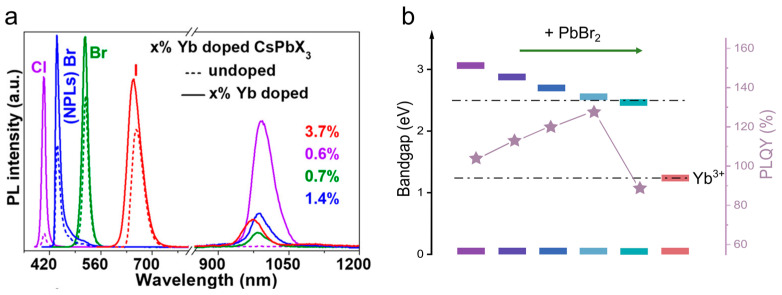
(**a**) PL spectra of undoped and Yb-doped CsPbX_3_ (X = Cl or I) pNCs and CsPbBr_3_ nanoplatelets. Dotted lines are for undoped samples, and solid lines of the same color are for Yb-doped samples. Adapted with permission from [30]. Copyright 2018 American Chemical Society. (**b**) Dependence of the bandgap and PL QY of Yb-doped CsPbCl_x_Br_3−x_ nanosheets on the variation of the Cl/Br halide content. The colors of the energy levels correspond to the color of the emission. Adapted from [16] with permission from Elsevier.

**Figure 5 nanomaterials-13-00744-f005:**
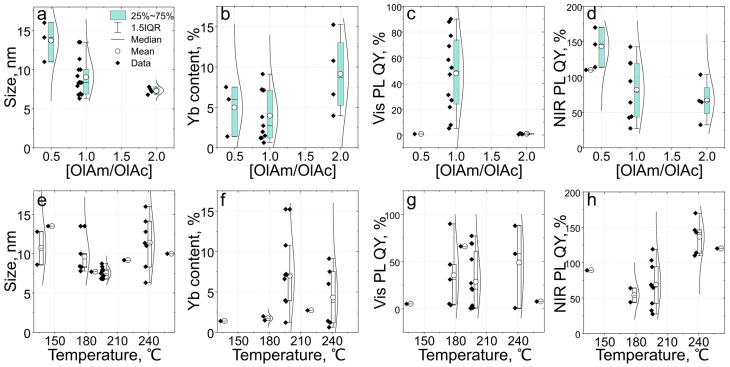
Dependency of the morphology and optical parameters of Yb-doped CsPbX3 pNCs on the ratio of oleylamine to oleic acid [OlAm/OlAc] (**a**–**d**) and the reaction temperature (**e**–**h**), (**a**,**e**) size, (**b**,**f**) Yb content, (**c**,**g**), Vis PL QY, (**d**,**h**) NIR PL QY. Data are taken from the refs. [13,15,17,23,26,31,43,44,46,55,56,57,58,59,60] and summarized in Table 1.

**Figure 6 nanomaterials-13-00744-f006:**
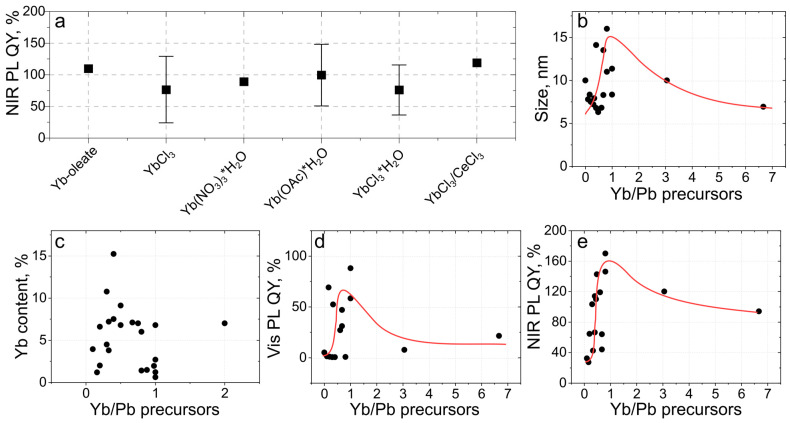
Dependencies of morphology and optical parameters of Yb-doped CsPbX_3_ pNCs on the molar ratio of Yb and Pb precursors. (**a**) Dependence of NIR PL QY on the type of Yb salt. Dependencies of size (**b**), Yb content (**c**), Vis PL QY (**d**), and NIR PL QY (**e**) on Yb-to-Pb precursor molar ratio (Yb/Pb precursors). Data are taken from the refs. [13,15,17,23,26,31,43,44,46,55,56,57,58,59,60] and summarized in Table 1.

**Table 1 nanomaterials-13-00744-t001:** Synthesis parameters and optical properties of Yb-doped pNCs.

Chemical Formula	Precursors	Yb Content, %	Yb/Pb Precursor Molar Ratio	Type of Ligands	OlAm/ OlAc Volume Ratio	T, °C	Reaction Time, sec	pNC Size, nm	Abs, nm	PL, nm	Vis PL QY, %	NIR PL QY, %	Ref.
Hot-injection method													
Yb:CsPbCl_3_	Cs-oleate, Pb-oleate, Yb-oleate, TMS-Cl	7.0	0.5	OlAm, OlAc	0.3	240	1	15	410	412	N/A	110	[15]
Yb:CsPbCl_3_	Cs_2_CO_3_, Pb(OAc)_2_, Yb(OAc)_3_, TMS-Cl	11.0–15.0	1.5	OlAm, OlAc	1.0	240	1	14	400	410	N/A	N/A	[17]
Yb:CsPbCl_x_Br_3−x_	Cs_2_CO_3_, Pb(OAc)_2_, Yb(OAc)_3_, TMS-Cl, TMS-Br	11.0–15.0	1.5	OlAm, OlAc	1.0	240	1	35	470	480	N/A	N/A	[17]
Yb:CsPbCl_x_Br_3−x_	Cs_2_CO_3_, Pb(OAc)_2_, Yb(OAc)_3_, TMS-Cl, TMS-Br	11.0–15.0	1.5	OlAm, OlAc	1.0	240	1	22	515	520	N/A	N/A	[17]
Yb:CsPbCl_3_	Cs-oleate, PbCl_2_, YbCl_3_	9.0	0.5	OlAm, OlAc	1.0	240	30	6.3	390	400	6.4	142.7	[23]
Yb:CsPbCl_3_	Cs-oleate, Pb(OAc)_2_, YbCl_3_	N/A	3.0	OlAm, OlAc	1.0	260	5	10	395	410	7.7	120.1	[26]
Yb:CsPbBr_3_@SiO_2_ *	Cs-oleate, PbBr_2_, YbCl_3_	1.5	0.7	OlAm, OlAc, APTES	1.0	180	60	13.5	490	484	31	64	[55]
Yb:CsPbBr_3_	Cs-oleate, PbBr_2_, YbCl_3_	2.0	0.7	OlAm, OlAc	1.0	180	60	8.3	490	476	47	44	[55]
Yb:CsPbCl_3_	CsOAc, Pb(OAc)_2_, Yb(OAc)_3_, TMS-Cl	7.5	0.4	OlAm, OlAc	0.5	240	1	14.1	405	410	N/A	114	[44]
Yb:CsPb(Cl_1−x_Br_x_)_3_ **	Yb:CsPbCl_3_ pNCs, TMS-Br	7.5	0.4	OlAm, OlAc	0.5	RT	12h	14.8	490	N/A	N/A	100	[44]
Yb:CsPbCl_3_	Cs-oleate, PbCl_2_, YbCl_3_	0.6	1.0	OlAm, OlAc, TOP	1.0	240	60	8.4	400	402	87.9	N/A	[56]
Yb:CsPbCl_3_	Cs-oleate, PbCl_2_, YbCl_3_	1.2	1.0	OlAm, OlAc, TOP	1.0	240	60	11.4	400	407	58.3	N/A	[56]
Yb:CsPbCl_3_	Cs-oleate, PbCl_2_, YbCl_3_	2.7	1.0	OlAm, OlAc, TOP	1.0	220	60	9.2	400	405	N/A	N/A	[56]
Yb:CsPbCl_3_	CsOAc, Pb(OAc)_2_, Yb(OAc)_3_, TMS-Cl	6.0	0.8	OlAm, OlAc	0.5	240	1	16	400	410	0.7	170	[46]
Yb:CsPbCl_3_:Mn(2.17%) ***	CsOAc, Pb(OAc)_2_, Yb(OAc)_3_, TMS-Cl, Mn(OAc)_2_	4.0	0.1	OlAm, OlAc	2.0	200	10	7.8	390	405	1.2	32.5	[57]
Yb:CsPbCl_3_:Mn(1.45%) ***	CsOAc, Pb(OAc)_2_, Yb(OAc)_3_, TMS-Cl, Mn(OAc)_2_	6.6	0.2	OlAm, OlAc	2.0	200	10	7.5	390	405	0.9	64.6	[57]
Yb:CsPbCl_3_:Mn(1.3%) ***	CsOAc, Pb(OAc)_2_, Yb(OAc)_3_, TMS-Cl, Mn(OAc)_2_	10.8	0.3	OlAm, OlAc	2.0	200	10	7.2	380	405	0.5	103.3	[57]
Yb:CsPbCl_3_:Mn(1.14%) ***	CsOAc, Pb(OAc)2, Yb(OAc)_3_, TMS-Cl, Mn(OAc)_2_	15.2	0.4	OlAm, OlAc	2.0	200	10	6.8	380	405	0.4	66.3	[57]
Yb:CsPbCl_3_	CsOAc, Pb(OAc)_2_, Yb(OAc)_3_, TMS-Cl	1.4	0.8	OlAm, OlAc	0.5	240	1	11	400	410	N/A	146	[58]
Yb:CsPb(Cl_1−x_Br_x_)_3_ **	Yb:CsPbCl_3_ pNCs, TMS-Br	1.4	0.8	OlAm, OlAc	0.5	RT	12h	N/A	480	495	N/A	N/A	[58]
Yb:CsPbCl_0.4_Br_2.6_	Cs-oleate, PbBr_2_, YbCl_3_	1.2	0.2	OlAm, OlAc	1.0	200	10	8.7	497	500	69.1	27.30	[13]
Yb:CsPbClBr_2_	Cs-oleate, PbBr_2_, YbCl_3_,	3.8	0.3	OlAm, OlAc	1.0	200	10	7.9	460	475	52.3	42.5	[13]
Yb:CsPbCl_1.5_Br_1.5_:Er(1.7%) ***	Cs-oleate, PbBr_2_, YbCl_3_, ErCl_3_	7.1	0.6	OlAm, OlAc	1.0	200	10	6.8	430	450	20.3	68.8	[13]
Yb:CsPbCl_1.5_Br_1.5_:Ce(2%) ***	Cs-oleate, PbBr_2_, YbCl_3_, CeCl_3_	7.1	0.6	OlAm, OlAc	1.0	200	10	6.8	430	450	27	119	[13]
Yb:CsPbCl_1.5_Br_1.5_	Cs-oleate, PbBr_2_, YbCl_3_	7.2	0.7	OlAm, OlAc	1.0	200	10	6.9	430	450	21.5	94.0	[13]
Post-synthetic treatment													
Yb:CsPbBr_1.5_Cl_1.5_	CsPbBr_1.5_Cl_1.5_ pNCs, Yb(OAc)_3_	5	N/A	OlAm, OlAc	N/A	RT	600	11.5	440	462	5.1	51	[43]
Yb:CsPbBr_1.5_Cl_1.5_	CsPbBr_1.5_Cl_1.5_ pNCs, Yb(OAc)_3_	8	N/A	OlAm, OlAc	N/A	RT	600	11	440	462	6.1	69	[43]
Yb:CsPbBr_1.5_Cl_1.5_	CsPbBr_1.5_Cl_1.5_ pNCs, Yb(OAc)_3_	10	N/A	OlAm, OlAc	N/A	RT	600	12	440	462	6.0	89	[43]
Yb:CsPbBr_1.5_Cl_1.5_	CsPbBr_1.5_Cl_1.5_ pNCs, Yb(OAc)_3_	20	N/A	OlAm, OlAc	N/A	RT	600	12	440	462	4.9	32	[43]
Yb:CsPbBr_1.5_Cl_1.5_	CsPbBr_1.5_Cl_1.5_ pNCs, Yb(OAc)_3_	30	N/A	OlAm, OlAc	N/A	RT	600	13.5	440	462	4.3	10	[43]
Yb:CsPbCl_3_	CsPbCl_3_ pNCs, Yb(NO_3_)_3_	0.6	N/A	OlAm, OlAc	N/A	RT	60	N/A	400	410	N/A	N/A	[57]
Yb:CsPbBr_3_	CsPbBr_3_ pNCs, Yb(NO_3_)_3_	0.7	N/A	OlAm, OlAc	N/A	RT	60	10	490	520	N/A	N/A	[57]
Yb:CsPbI_3_	CsPbI_3_ pNCs, Yb(NO_3_)_3_	3.7	N/A	OlAm, OlAc	N/A	RT	60	N/A	630	660	N/A	N/A	[57]
Yb:CsPbBr_3_ NPLs	CsPbBr_3_ pNPLs, Yb(NO_3_)_3_	1.4	N/A	OlAm, OlAc	N/A	RT	60	N/A	430	440	N/A	N/A	[57]
Yb:CsPbCl_3_	CsPbCl_3_ pNCs, YbCl_3_	N/A	N/A	OlAm, OlAc	N/A	RT	12h	N/A	400	405	2	N/A	[31]
Yb:CsPbBr_3_	CsPbBr_3_ pNCs, YbCl_3_	N/A	N/A	OlAm, OlAc	N/A	RT	12h	6.6	410	415	2	5	[31]

Notes: OAc—acetate, TMS—trimethylsilyl, OlAm—oleylamine, OlAc—oleic acid, APTES—(3-aminopropyl) triethoxysilane, TOP—trioctyl phosphate, N/A—information is not available, RT—room temperature, reaction time of 12 h—‘overnight’, NCs—nanocrystals, NPLs—nanoplatelets; * pNCs were synthesized in presence of APTES and are covered by silica; ** halide exchange in Yb-doped pNCs; *** co-doped CsPbX_3_ pNCs, in brackets the content of the second dopant is given.

## Data Availability

The database is available by link: https://github.com/VladislavNexby/ML (accessed on 15 February 2023).

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
