# Peer review of "Ytterbium-Doped Lead–Halide Perovskite Nanocrystals: Synthesis, Near-Infrared Emission, and Open-Source Machine Learning Model for Prediction of Optical Properties"

_nanomaterials, 2023, doi:10.3390/nano13040744_

Round 1

Reviewer 1 Report

Nanomaterials: Review report

Comments:
            The authors have well organized the review article “Ytterbium-doped lead-halide perovskite nanocrystals: synthesis, near-infrared emission, and open-source machine learning model for prediction of optical properties”. After thorough review, I felt this article deserves a fine space in the journal “Nanomaterials”.

Introduction: The background, motivation and literature review has been good and organized well.

 Synthesis: Methodology of synthesis of Pb-halide perovskites has been mentioned clearly with appropriate diagrams.

 Different properties and their discussions: Clearly demonstrated morphology, optical properties and dependence of the properties of Yb-doped lead-halide pNCs on synthesis parameters along with huge data in the table for comparison as well as machine learning model for prediction of optical properties.

             Conclusions: OK, but still a comment: Could have mentioned the flaws in Pb-based perovskites along with remedies.

The results deserve to be published in this journal even in the present form.

Author Response

Response:

We appreciate the positive recommendation from the reviewer. The conclusion section has been revised according to the comment.

Page 13: “6. Summary and Outlook

To date, the perovskite <…> However, these materials are still far from industrial use since they have several drawbacks such as fast oxidation and poor photostability in the UV spectral region. An important target is the development of simple, scalable, and reliable synthetic procedures for producing perovskite materials with well-controlled properties such as bright emission and tunable energy structure. The doping of pNCs with lanthanides, along with expanding the spectral range of their optical transitions to NIR, allows us to overcome some existing issues of perovskite materials, through improving their environmental stability, decreasing the intrinsic defects and, hence, improving the emissive properties in the visible spectral region. <…> The optimization of synthetic procedures may lead to a better performance of perovskite-based devices. Obviously, extended use of the existing and emerging instruments of data analysis and machine learning will play an important role in improving synthetic protocols and developing novel lanthanide doped perovskite materials.

Based on the literature data, an open-source code for approximation of NIR PL QY dependence on synthetic parameters was proposed. This model can solve the direct problem of predicting the values of the PL QY in the NIR region of the spectrum from the synthesis parameters of CsPbCl3 pNCs and it can be considered as a base for further improvements, which may be accomplished in several possible ways:

i) development of standardized template for data collection and further expansion of the datasets based on available experimental data will improve the accuracy of the model prediction;

ii) other machine learning models, regularization techniques and other methods of preprocessing can be employed and compared for better model performance;

iii) improvement of mathematical models may include such functions as prediction of maximal values, sorting by type of precursors and/or resulting materials, graphical representation of derived dependencies of optical characteristics on synthesis parameters, and many more.

Properly performed data analysis will facilitate optimization of the existing synthesis methods together with a possibility to predict optical properties of doped perovskite materials by setting the synthesis parameters. Thus, employment of machine learning methods offers a great potential for improvement of the doped perovskite materials in particular and plenty of novel nanostructured materials in general.”

Reviewer 2 Report

Several methods of perovskite nanocrystal synthesis and doping are described in depth. In particular, it is judged that it will be of great help to the synthesis of perovskite in the future by well synthesizing the results of various synthesis conditions based on machine learning. The content is logically well written overall, and it seems that only a minor spell check is required

Author Response

Response:

We appreciate the positive recommendation from the reviewer. The text has been checked and revised according to the comment.

Reviewer 3 Report

This review article provides the recent progress in the ytterbium-doped metal halide perovskites with a particular focus on the synthesis parameters and how these parameters affect the material properties, such as morphologies and optical properties. In addition, an open-source database has been initiated for future prediction of the synthetic protocols for these materials. This review has a very focused summary on ytterbium-doped halide perovskites. I would recommend to publish after some minor revisions.

1.       Format of the writing should be improved. For example, spaces should be used/removed for various places throughout the paper (Lin275, Page 2; ).  Page 4, line 120, “several min” should be changed to “several minutes”. Page 11, line 266, lower case should be used for the chemical formula of CsPbX3.

2.       Some relevant papers are recommended to be cited in the introduction section, highlighting the significance of the doping strategy in metal halide perovskite research field, such as Nano Energy, 2020, 69 104392, doi: 10.1016/j.nanoen.2019.104392; Advanced Functional Materials, 2022, 32 (52) 2208077, 1-11. doi: 10.1002/adfm.202208077;

3.       The applications of the ytterbium-doped halide perovskites are too brief. It`s recommended that more discussion on the published work on applications of doped halide perovskites can be added.

4.       Research problems in this field and the future research directions are very important part of a review paper. Discussion on these sections are needed in the conclusion section.

Author Response

Reviewer #3

This review article provides the recent progress in the ytterbium-doped metal halide perovskites with a particular focus on the synthesis parameters and how these parameters affect the material properties, such as morphologies and optical properties. In addition, an open-source database has been initiated for future prediction of the synthetic protocols for these materials. This review has a very focused summary on ytterbium-doped halide perovskites. I would recommend to publish after some minor revisions.

Response:

We appreciate the positive recommendation from the reviewer.

  1. Format of the writing should be improved. For example, spaces should be used/removed for various places throughout the paper (Lin275, Page 2; ).  Page 4, line 120, “several min” should be changed to “several minutes”. Page 11, line 266, lower case should be used for the chemical formula of CsPbX3.

Response:

The text has been checked and revised according to the comment.

  1. Some relevant papers are recommended to be cited in the introduction section, highlighting the significance of the doping strategy in metal halide perovskite research field, such as Nano Energy, 2020, 69 104392, doi: 10.1016/j.nanoen.2019.104392; Advanced Functional Materials, 2022, 32 (52) 2208077, 1-11. doi: 10.1002/adfm.202208077;

Response:

We thank the reviewer for this valuable comment. These papers are now included in the reference list as #7 and #10.

Page 1: “This fascinating luminous efficiency makes them promising materials for light-emission and light-conversion applications, including light-emitting diodes, lasers, luminescence solar concentrators, and planar solar cells. [4-7]”.

Page 2: “Modification of perovskites with dual cations (divalent Cd2+ and monovalent K+) was shown to be very efficient in reducing the defects, immobilizing the halide anions, and preventing ion loss from perovskite during post-annealing process, which improved power conversion efficiency of perovskite solar cells [10]”.

  1. The applications of the ytterbium-doped halide perovskites are too brief. It`s recommended that more discussion on the published work on applications of doped halide perovskites can be added.

Response:

The section on applications of Yb-doped perovskites has been extended in the Introduction section.

Page 2: “In particular, Zhou et al. deposited a layer of Yb3+-Ce3+ co-doped CsPbCl1.5Br1.5 pNCs on top of a commercial single-crystal silicon solar cell [13]. Optimization of the thickness of perovskite layer provided both high transparency in the visible and NIR regions and effective capture of the ultraviolet light which was converted to the Yb3+-related NIR radiation. As a result, the power conversion efficiency increased from 18.1% to 21.5%. Luo et al. demonstrated a concept of a quantum-cutting luminescent solar concentrator with Yb3+-doped CsPbCl3 pNC embedded into a polymethyl methacrylate matrix [14]. The prototype of a solar concentrator with the dimensions of 5 cm × 5 cm × 0.2 cm demonstrated an internal quantum efficiency of 118.1±6.7%, 2-fold higher than previous records. Huang et al. deposited Yb3+-doped CsPbCl3 onto a UV commercial chip to build a NIR-emitting light-emitting diode [15]. The device demonstrated peak external quantum efficiency of 2% and good operational stability. Sun et al. used laterally assembled Yb3+-doped CsPbClBr2 nanosheets to fabricate a dual-band sensitive photodetector [16]. The device demonstrated the responsivity of 1.96 A·W-1 (0.12 mA·W-1) and detectivity of 5 × 1012 (2.15 × 109) Jones at 440 (980) nm wavelength”.

  1. Research problems in this field and the future research directions are very important part of a review paper. Discussion on these sections are needed in the conclusion section.

Response:

The conclusion section has been extended according to the comment.

Page 13: “6. Summary and Outlook

To date, the perovskite <…> However, these materials are still far from industrial use since they have several drawbacks such as fast oxidation and poor photostability in the UV spectral region. An important target is the development of simple, scalable, and reliable synthetic procedures for producing perovskite materials with well-controlled properties such as bright emission and tunable energy structure. The doping of pNCs with lanthanides, along with expanding the spectral range of their optical transitions to NIR, allows us to overcome some existing issues of perovskite materials, through improving their environmental stability, decreasing the intrinsic defects and, hence, improving the emissive properties in the visible spectral region. <…> The optimization of synthetic procedures may lead to a better performance of perovskite-based devices. Obviously, extended use of the existing and emerging instruments of data analysis and machine learning will play an important role in improving synthetic protocols and developing novel lanthanide doped perovskite materials.

Based on the literature data, an open-source code for approximation of NIR PL QY dependence on synthetic parameters was proposed. This model can solve the direct problem of predicting the values of the PL QY in the NIR region of the spectrum from the synthesis parameters of CsPbCl3 pNCs and it can be considered as a base for further improvements, which may be accomplished in several possible ways:

i) development of standardized template for data collection and further expansion of the datasets based on available experimental data will improve the accuracy of the model prediction;

ii) other machine learning models, regularization techniques and other methods of preprocessing can be employed and compared for better model performance;

iii) improvement of mathematical models may include such functions as prediction of maximal values, sorting by type of precursors and/or resulting materials, graphical representation of derived dependencies of optical characteristics on synthesis parameters, and many more.

Properly performed data analysis will facilitate optimization of the existing synthesis methods together with a possibility to predict optical properties of doped perovskite materials by setting the synthesis parameters. Thus, employment of machine learning methods offers a great potential for improvement of the doped perovskite materials in particular and plenty of novel nanostructured materials in general.”